# Improving Biomedical Abstractive Summarisation with Knowledge Aggregation from Citation Papers

**Chen Tang[1], Shun Wang[2], Tomas Goldsack[2] and Chenghua Lin[2,3]***

[1]Department of Computer Science, The University of Surrey, UK
[2]Department of Computer Science, The University of Sheffield, UK
[3]Department of Computer Science, The University of Manchester, UK
chen.tang@surrey.ac.uk, chenghua.lin@manchester.ac.uk
{swang209, tgoldsack1}@sheffield.ac.uk

## Abstract

Abstracts derived from biomedical literature possess distinct domain-specific characteristics, including specialised writing styles and biomedical terminologies, which necessitate a deep understanding of the related literature. As a result, existing language models struggle to generate technical summaries that are on par with those produced by biomedical experts, given the absence of domain-specific background knowledge. This paper aims to enhance the performance of language models in biomedical abstractive summarisation by aggregating knowledge from external papers cited within the source article. We propose a novel attention-based citation aggregation model that integrates domain-specific knowledge from citation papers, allowing neural networks to generate summaries by leveraging both the paper content and relevant knowledge from citation papers. Furthermore, we construct and release a large-scale biomedical summarisation dataset that serves as a foundation for our research. Extensive experiments demonstrate that our model outperforms state-of-the-art approaches and achieves substantial improvements in abstractive biomedical text summarisation.

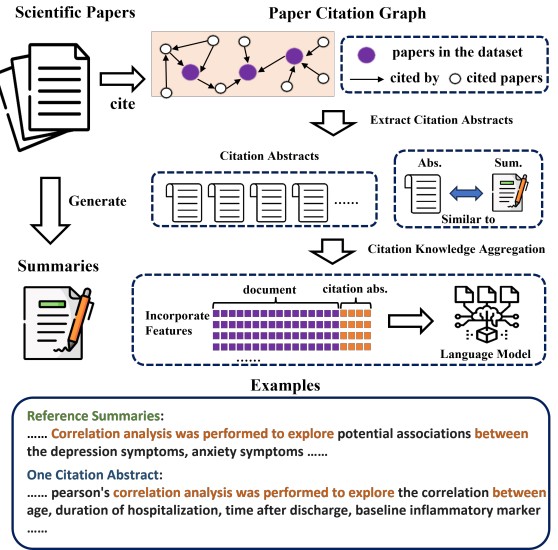

Figure 1: The overview of our proposed citation knowledge aggregation framework. In this framework, language models are trained to incorporate features from both the content of the main paper and the abstracts of its cited papers. The rationale behind this approach is that the cited papers often share relevant research backgrounds, terminologies and writing styles, which can be used as a good template for the summary generation.

## 1 Introduction

Biomedical text summarisation plays a pivotal role in facilitating the comprehension of the vast and constantly expanding body of biomedical literature (Xie et al., 2022), which poses a significant challenge for clinicians and domain experts who strive to remain well-informed in their respective fields. To address this challenge, the generation of high-quality summaries from the extensive corpus of biomedical literature holds immense potential in supporting research and advancements within the biomedical domain (DeYoung et al., 2021).

One of the key challenges in biomedical natural language generation (NLG) lies in effectively handling domain-specific terminologies that are prevalent in biomedical texts. Consequently, a plethora of research studies have been conducted with a primary focus on enhancing language quality by better integrating domain-specific knowledge in the biomedicine domain (Sotudeh Gharebagh et al., 2020; Tangsali et al., 2022; An et al., 2021; Tang et al., 2023b) However, most prior works have predominantly attempted to incorporate knowledge by leveraging additional annotations within the paper content. These annotations include frequent items (Givchi et al., 2022), named entities (Schulze and Neves, 2016; Peng et al., 2021), entity relations (Shang et al., 2011), as well as external knowledge systems such as biomedical ontologies (Chandu et al., 2017) and external terminology

---
*Corresponding author.

searching tools (Gigioli et al., 2018). Surprisingly, the inclusion of external knowledge derived from citation papers has been rarely explored in previous biomedical studies. Existing corpora for biomedical text summarisation are typically constructed in a manner that models solely rely on the source article when generating a summary. However, as shown in Figure 1, there exists strong connections among papers in the citation network with shared research backgrounds, terminologies, and abstract styles, which will be a useful source of knowledge for improving biomedical abstractive summarisation but not captured in existing datasets.

To address this gap in the existing biomedical summarisation dataset, we construct a novel biomedical summarisation dataset utilising an open-source biomedical literature corpus provided by the Allen Institute[1]. During the dataset construction process, we applied rigorous filtering criteria to eliminate low-quality samples. Specifically, we discarded samples with an insufficient number of citations (less than three distinct citations), as well as unqualified papers whose unique identifiers (UIDs) or citation UIDs were inaccessible within the corpus. Additionally, we designed heuristic rules to select and transform the unstructured raw data corpus into a structured dataset in JsonL format. The final dataset comprises over 10,000 instances, with each instance having an average of 16 citations. To the best of our knowledge, this is the largest biomedical literature dataset[2] specifically tailored for citation paper-enhanced biomedical text summarisation. Furthermore, we provide the corresponding methods for collecting the citation network, including cited papers and their associations.

Facilitated by our biomedical summarisation dataset, we further propose a novel approach to biomedical document summarisation whereby we enhance neural models with external domain-specific knowledge in the form of the abstracts of cited papers. Accordingly, we introduce an attention-based network (Vaswani et al., 2017) that dynamically aggregates features extracted from the citation abstracts with the encoded content features of the main paper. This aggregation is achieved by applying attention mechanisms to the associated abstracts of all cited papers, which provides the subsequent summary decoding process with additional features derived from abstracts of the citation papers. Within this framework, the base language model can effectively leverage both the features of the main paper and the additional domain-specific knowledge obtained from cited papers. Consequently, this integration leads to enhanced performance in text summarisation. Extensive experiments demonstrate that our model outperforms state-of-the-art baselines in abstractive biomedical text summarisation. We also conducted an in-depth quantitative analysis to verify the performance gain obtained by our attention-based citation knowledge enhancement framework[3]. Our contributions are summarised as follows:

- We construct a large-scale biomedical literature dataset, which can be used for enhancing biomedical text summarisation with the extracted external knowledge from cited papers.

- We propose a novel framework that can effectively leverage citation papers to enhance the performance of large-scale language models on abstractive summarisation of biomedical literature.

- We conduct extensive experiments to evaluate the effectiveness of our proposed framework, including comparisons with SOTA models and an in-depth analysis of the performance gain achieved by aggregating different quantities of citations.

## 2 Related Work

In recent years, a variety of large-scale pre-trained models (PLMs), such as **BART** (Lewis et al., 2019); **T5** (Raffel et al., 2020); **GPT-2** (Radford et al., 2019), have demonstrated remarkable performance improvements across various tasks (Loakman et al., 2023; Zhang et al., 2023; Zhao et al., 2023; Tang et al., 2022b) in the Natural Language Generation (NLG) Domain. These PLMs have also been widely applied to biomedical text summarisation. These models, e.g. BioBERT (Lee et al., 2020) and BioBART (Yuan et al., 2022), have achieved remarkable performance by training on extensive biomedical literature corpora, such as Pubmed[4] and MIMIC-III[5]. However, certain high-level knowledge, e.g., the understanding of medical terminologies, cannot be adequately captured solely

---

[1] https://allenai.org/data/cord-19

[2] The sole viable dataset we have identified is The Text Analysis Conference (TAC) 2014 Biomedical Summarization track (Cohan et al., 2014) comprising mere 313 instances.

[3] Our code and data resources is accessible at https://github.com/tangg555/biomed-sum.

[4] https://pubmed.ncbi.nlm.nih.gov/

[5] https://physionet.org/content/mimiciii/1.4/

through the implicit modeling of word probabilities. To address this limitation, the improvement of biomedical background knowledge understanding is able to necessitate the integration of additional knowledge systems, such as conceptual ontologies. These ontologies explicitly model representations of domain-specific knowledge learned by neural networks. Recent studies have proposed incorporating biomedical knowledge, including terminologies (Tang et al., 2023b)) and concepts (Chandu et al., 2017), to enhance the performance of these language models and bridge the gap between language understanding and specialized biomedical knowledge. Indeed, several notable works have focused on enhancing summarisation through citations in the open domain, such as An et al. (2021) and Yasunaga et al. (2019). However, it is important to highlight that the progress of language models in the biomedical domain has been hindered by the limited availability of datasets and resources. This scarcity has impeded the further advancement and improvement of pre-trained language models (PLMs) specifically tailored for biomedical applications. In this study, we require a dataset that contains retrievable citation papers, making traditional raw data corpora such as Pubmed and MIMIC-III inadequate. To date, the sole public dataset we could find is the Text Analysis Conference (TAC) 2014 Biomedical Summarization track (Cohan et al., 2014). However, this dataset is limited in size comprising merely 313 instances, and is somewhat outdated. Therefore, we construct a novel dataset for investigating biomedical citation-enhanced summarisation.

## 3 Dataset Construction

### 3.1 Construction Process

In order to create a dataset containing biomedical literature and its associated citations, we process a semi-structured raw corpus [6] released by Allen Institute. We refer to this dataset as BioCiteDB throughout the paper. The construction process of the dataset is outlined in algorithm 1, where $C$ represents the raw corpus, and $D$ represents the processed dataset. To ensure the quality and relevance of the data, the selected papers have to meet the following requirements: (1) The papers must include

---

[6]We select the latest version of CORD-19 Historical Releases (2022-06-02 18.7 GB), which can be accessed at https://ai2-semanticscholar-cord-19.s3-us-west-2.amazonaws.com/historical_releases.html.

---

**Algorithm 1:** Construction of BioCiteDB

**Input:** Samples $c_i \in C$; Citation limit $R$
**Output:** Json objects $d_i \in D$
1  **Initialise** $D \leftarrow \varnothing$
2  **foreach** $c_i$ in $C$ **do**
3      Initialise object $d_i$ with $c_i$
4      retrieve files $f_j$ to a queue $q$
5      Initialise object $p_i$
6      **foreach** $f_j$ in $q$ **do**
7          **if** $f_j$ missing elements **then**
8              break
9          **end**
10         extract distinct citations $r_n \in f_i$
11         **if** $|r_n| >= R$ **then**
12             $r_n.uid \rightarrow p_i.citations$
13             extend $d_i$ with $p_i$
14             break
15         **end**
16     **end**
17     $D$ append $d_i$
18 **end**
19 **foreach** $d_i$ in $D$ **do**
20     **foreach** $r_n$ in $d_i.citations$ **do**
21         **if** $r_n \notin D$ **then**
22             exclude $r_n$
23         **end**
24     **end**
25 **end**

| Datasets | Train | Val | Test |
|---|---|---|---|
| **# Samples** | 9144 | 1143 | 1143 |
| **# Papers** | 18194 | 4762 | 4618 |
| **Avg. # Distinct Citations of Doc** | 6.28 | 6.23 | 6.26 |
| **Avg. # Total Citations of Doc** | 16.56 | 16.85 | 16.96 |
| **Avg. # Chunks in Doc** | 37.33 | 37.33 | 37.13 |
| **Avg. # Sentences in Doc** | 529.94 | 526.8 | 529.11 |
| **Avg. # Words in Doc** | 9858.09 | 9901.96 | 9907.31 |
| **Avg. # Sentences in Summary** | 13.96 | 13.98 | 13.69 |
| **Avg. # Words in Summary** | 220.25 | 222.04 | 217.28 |

Table 1: Data statistics of Biomed Ref dataset. Doc is the abbr. of Document, and Intro is the abbr. of introduction. Chunks are split by section and subsections in a paper.

the "Introduction" section, as it is considered the most crucial part for generating abstracts; (2) The papers must have at least three distinct citations to ensure the quality of curated data; (3) The essential elements of the papers, including UID (Pubmed id), Title, Abstract, Sections, and Citations, must be accessible within the raw corpus. As a result of this construction process, the dataset $D$ comprises structured data in JsonL[7] format, with each sample representing an individual paper.

### 3.2 Data Statistics

The statistical analysis of our processed dataset is presented in Table 1. Additionally, the distribution

---

[7]https://manifold.net/doc/mfd9/jsonl.htm.

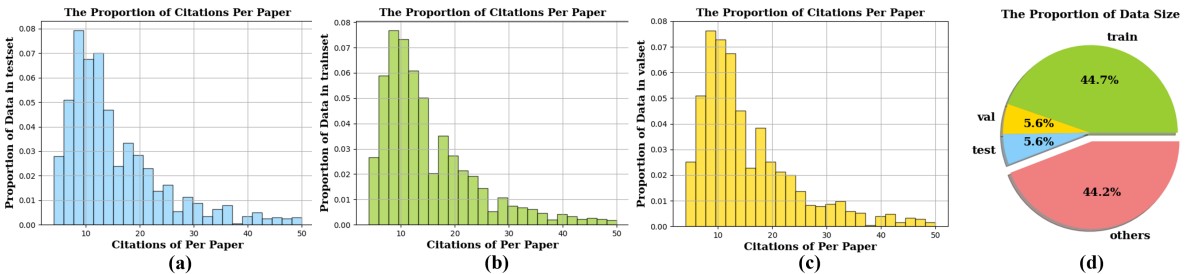

Figure 2: A visulisation of the distribution of citations per paper and the data size associated with each split. To avoid clutter and maintain clarity, we have excluded papers with over 50 citations from the visualization, as they constitute a relatively small proportion of the dataset and fall into the long tail category. Some papers in our corpus are only used as citations other than the papers in data splits, so we categorise them as "others".

of citations per paper is visualized in Figure 2 (a), (b), and (c), while the proportions of data size are depicted in Figure 2 (d) of the same figure. The results obtained from both the statistical analysis and visual representations in Table 1 and Figure 2 both validate the data quality of the constructed dataset, thus indicating the effectiveness of our data construction process and the consistency of the dataset splits. This validation supports the notion that training and inference tasks conducted on this dataset can be regarded as fair and reliable.

---

**Algorithm 2:** Extracting Citation Graph $G$.

**Input:** $d_i \in D$; $hop_{max}$; $N$
**Output:** The set of related papers $P$

1 **Initialise** current hop $hop_n = 0$; a double-ended queue $DQ \leftarrow (d_i.uid, hop_n)$; a queue recording visited nodes $VQ$;
2 **while** $s^i$ in $S$ **do**
3    pop $uid$ and $hop_n$ from $DQ$
4    **if** $hop_n > hop_{max}$ **then**
5      return $P$
6    **end**
7    $P \leftarrow (uid, hop_n)$
8    **if** $|P| > N$ **then**
9      return $P$
10    **end**
11    get $d_j$ by $uid$
12    $VQ \leftarrow uid$
13    **foreach** $r_n \in d_j.citations$ **do**
14      **if** $r_n.uid \notin vq$ and $r_n \in D$ **then**
15        $P \leftarrow r_n.uid$ and $VQ \leftarrow r_n.uid$
16      **end**
17    **end**
18 **end**

---

### 3.3 Extract Citation Graph

Scientific papers are intricately connected through citation relationships, forming a network of interconnected nodes. This citation graph provides valuable insights into the relatedness of papers. In order to retrieve relevant papers within this cita-

tion graph, we propose an algorithm outlined in algorithm 2. $hop_{max}$ defines the maximum number of hops between papers that the algorithm can traverse, and $N$ specifies the maximum number of retrieved papers at each hop. As output, $P$ represents papers as nodes, while citation relationships are represented as edges in the network. Due to the high computational cost of processing long documents for summarisation, we set $hop_{max}$ to 1 and $neighbor_{max}$ to 12, taking into account the limitations of our available computing resources. However, it is worth noting that the attention-based citation aggregation module can be extended to incorporate Graph Attention Networks (Zhou et al., 2020), which have the capability to integrate multi-layer citation graphs (Zhang et al., 2023).

## 4 Methodology

As illustrated in Figure 3, our proposed framework is designed to enhance the performance of the base language model by leveraging the collective knowledge from a set of citation papers. For our experiments, we select BART (Lewis et al., 2019), a widely-used summarization model that has demonstrated promising results in the biomedical domain (Goldsack et al., 2022, 2023), as the base model. In this study, we adopt a strategy where we concatenate the abstracts of the citation papers with the input document to form the model's input. This approach is motivated by the goal of enabling the model to capture and emulate the writing style present in relevant papers. By incorporating this additional information, we aim to improve the model's ability to generate high-quality summaries that align with the conventions and patterns observed in the domain-specific literature.

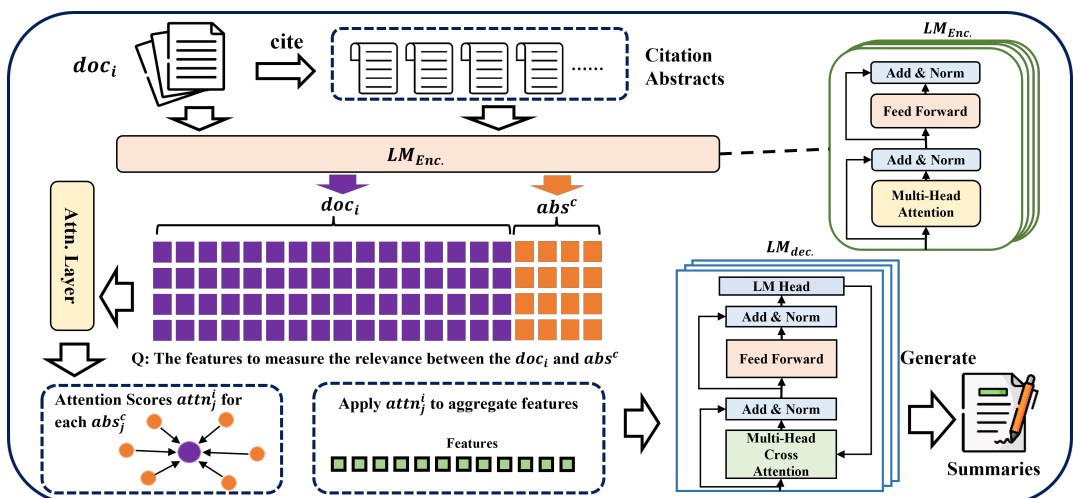

Figure 3: The illustration of our proposed framework. **doc** is the abbr. of the input document referring to the paper content. **abs** is the abbr. of the abstracts and $abs_c$ denotes the abstracts of citation papers.

## 4.1 Task Definition

The task is formulated as follows: Given a paper document $d_i \in D$ as the input, where $D$ represents the paper corpus, and $d_i$ denotes the $i$-th paper. In addition, the citations papers $D^c = \{d_1^c, d_2^c, ..., d_k^c\}$ are also provided as the input. The abstracts of $d_k^c \in D^c$ are denoted $abs_k^c$. Either $d_i$ or $d_j^c$ consists of a sequence of words represented as $X = \{x_1, x_2, ..., x_t\}$ where $x_t$ denotes $t$-th word in $X$. The goal is to generate a summary $Y = \{y_1, y_2, ..., y_t\}$ by modeling the conditional probability distribution $P(Y|X \in d_i, X \in D^c)$.

## 4.2 Knowledge Aggregation from Citations

**Input** At the initial stage, both the input document $d_i$ and its retrieved $N$ citation abstracts $abs^c$ are concatenated and encoded by language models. Byte-Pair Encoding (Radford et al., 2019) is implemented in the transformation from text into fixed word embeddings:

$$E_{doc} = \text{LM}_{emb}([Tok^{CLS}, x_t \in d_i]) \quad (1)$$

$$E_{abs_j^c} = \text{LM}_{emb}([Tok^{ABS}, x_t \in abs_j^c]) \quad (2)$$

$$E_{Q_j} = \text{concat}(E_{doc}, E_{abs_j^c}) \quad (3)$$

where $LM_{emb}$ represents the module responsible for tokenising and converting words into sub-word embeddings. $Tok^{CLS}$ is a special token that signifies the global context tag in the input text. $Tok^{ABS}$ is a special token used to indicate the separation between the input document and the cited abstracts. $E_{Q_j}$ denotes the embeddings generated for the $j$-th ($j \in [1, N]$) document abstract pair.

**Encoding** In order to capture the relevance of each cited abstract, we employ an attention mechanism to measure the importance of $d_i$ with respect to $abs_j^c$. The attention score is denoted as $attn_j^i$, and the process of aggregating knowledge is illustrated as follows:

$$E_Q = \text{concat}([E_{Q_1}, ..., E_{Q_N}]) \quad (4)$$

$$Q = \text{LM}_{enc}(E_Q), Q \in \mathbb{R}^{N \times L \times M} \quad (5)$$

where $E_Q$ denotes the matrix of embeddings for all composed $Q_j$, and it is encoded by the language model encoder to generate the encoded features $Q$.

$$Q^{CLS} = \text{First\_Pool}(Q), Q^{CLS} \in \mathbb{R}^{N \times M} \quad (6)$$

$$Attn\_logits = Q^{CLS}W^Q, Attn \in \mathbb{R}^{N \times 1} \quad (7)$$

$$Attn = \text{softmax}(Attn), Attn \in \mathbb{R}^{N \times 1} \quad (8)$$

$$F = A^T Q, F \in \mathbb{R}^{L \times M} \quad (9)$$

In the above equations, $First\_Pool$ collects features that represent the global context of the input $d_i$ and $abs_j^c$ pairs. As the hidden states of the neural encoder, $Q$ incorporates features from both the documents and the abstracts. Therefore, the representations of the first position in $Q$ (represented as $Q^{CLS}$) correspond to the global context token $Tok^{CLS}$. The attention logits matrix is obtained by applying a trainable parameter $W^Q \in \mathbb{R}^{M \times 1}$ to the features of $Q^{CLS}$. After applying the $softmax$ function for normalization, $Attn$ represents the importance of the input features and is used to reweight the original encoded features $Q$, resulting in the final features $F$.

## 4.3 Summary Generation

In line with other abstractive summarization systems, we employ an auto-regressive decoder to generate summary tokens $y_t$ in a sequential manner. The process is described as follows:

$$H_t = \text{Decoder}(y_{<t}, F) \tag{10}$$

$$P(y_t|y_{<t}, X) = \text{softmax}(H_t W^D) \tag{11}$$

$$y_t \xleftarrow{\text{sampling}} P(y_t|y_{<t}, F) \tag{12}$$

where $t$ represents the current time step. $X$ corresponds to the input, consisting of the words from $d_i$ and $abs_1^c, ..., abs_j^c$, provided to the neural model. $H_t$ refers to the hidden state of the decoder module at time step $t$. This state is computed by the language models using the infused features $F$, which encapsulate the information from the input document and its cited abstracts, along with the previously predicted tokens $y_{<t}$. $W^D$ denotes a trainable parameter, and $P(y_t|y_{<t}, F)$ represents the probability distribution over the vocabulary, which includes special tokens. Employing a sampling strategy, such as $argmax$, we obtain the predicted token $y_t$.

## 4.4 Training and Inference

Finally, as shown in Figure 3, the neural model is trained to fit on the citation-enhanced training set by the following objective function:

$$\mathcal{L} = -\frac{1}{N} \sum_{t=1}^{N} \log P(y_t|y_{<t}, X) \tag{13}$$

where $\mathcal{L}$ is the cross-entropy loss employed to train the model in modeling the conditional probabilities over the token sequence $P(y_t|y_{<t}, F)$. By minimizing $\mathcal{L}$, the language model learns to predict the referenced abstract corresponding to the input document.

## 5 Experiment

### 5.1 Experimental Setup

**Baselines** We include a range of competitive PLM models as our baselines. We also provide the results of two rule-based systems, namely LEAD-3 and ORACLE, which serve as benchmarks representing the upper and lower bounds of model performance. **LEAD-3** extracts the first 3 sentences from the input as the summary, which can be considered as the lower bound of the performance. **ORACLE** select sentences from the input document and compose a summary with the highest score[8], which is the upper bound of extractive summarisation systems. The PLM models serve as baselines for abstractive biomedical summarisation. The **Long-Document Transformer** (**LED**) is a Transformer-based models which are able to process long sequences due to their self-attention operation (Beltagy et al., 2020). **PEGASUS** (Zhang et al., 2020) is pre-training large Transformer-based encoder-decoder models on massive text corpora with a new self-supervised objective, which is tailored for abstractive text summarisation. **BART** (Lewis et al., 2019) is a widely used PLM model based on a denoising autoencoder that has proved effective for long text generation tasks. **Pubmed-X** refers to several PLM-based baselines that have been pre-trained on a large-scale biomedical literature corpus Pubmed (Cohan et al., 2018) (the size of the dataset is 215k), where **X** denotes the name of a PLM. Additionally, we include **ChatGPT** for comparison. However, as it is close-source and very expensive for training, we were unable to use ChatGPT as the base language model to fine-tune on our dataset. Instead, we compare the outputs of **ChatGPT** in a zero-shot setting.

**Evaluation Metrics** In the domain of text summarisation (Sun et al., 2021; Tang et al., 2022a; Xie et al., 2022), ROUGE (Lin, 2004) is the most used metric for the evaluation of generated summaries. For evaluating the quality of the generated summaries, we implement the ROUGE metric with the python package of *rouge_score*. Specifically, we report the unigram and bigram overlaps (*ROUGE-1* and *ROUGE-2*, respectively) between the generated summaries and the reference (golden) summaries. Additionally, we include the longest common subsequence (*ROUGE-L*) metric to evaluate the fluency of the generated summaries. For each ROUGE metric, we provide fine-grained measurements of precision, recall, and F-values, offering a comprehensive assessment of the summarisation performance.

In addition to the ROUGE metric, we conduct an extensive automatic evaluation utilising a broader range of evaluation metrics. Specifically, we em-

---

[8]In this study, the score referenced by ORACLE is calculated as the mean value of the ROUGE-1, ROUGE-2, and ROUGE-L scores.

| Models | PPL↓ | ROUGE-1↑ | | | ROUGE-2↑ | | | ROUGE-L↑ | | |
|---|---|---|---|---|---|---|---|---|---|---|
| | | Precision | Recall | F-value | Precision | Recall | F-value | Precision | Recall | F-value |
| **LEAD-3** | - | 0.5512 | 0.1838 | 0.2645 | 0.2039 | 0.0647 | 0.0941 | 0.4979 | 0.1646 | 0.2374 |
| **ORACLE** | - | 0.5669 | 0.4121 | 0.4676 | 0.2478 | 0.1764 | 0.2015 | 0.5195 | 0.3762 | 0.4276 |
| **ChatGPT** | - | 0.4965 | 0.3899 | 0.4242 | 0.1720 | 0.1358 | 0.1471 | 0.4551 | 0.3564 | 0.3880 |
| **LED** | 11.36 | 0.3250 | 0.3129 | 0.3150 | 0.0940 | 0.0909 | 0.0912 | 0.2934 | 0.2822 | 0.2844 |
| **PEGASUS** | 12.54 | 0.3063 | 0.3036 | 0.3003 | 0.0837 | 0.0833 | 0.0821 | 0.2715 | 0.2681 | 0.2657 |
| **BART** | 10.89 | 0.3581 | 0.2963 | 0.3171 | 0.0862 | 0.0709 | 0.0760 | 0.3276 | 0.2719 | 0.2907 |
| **Pubmed-LED** | 11.01 | 0.3462 | 0.3394 | 0.3395 | 0.0877 | 0.0860 | 0.0859 | 0.3109 | 0.3044 | 0.3047 |
| **Pubmed-PEGASUS** | 18.27 | 0.3806 | 0.2463 | 0.2926 | 0.0980 | 0.0635 | 0.0752 | 0.3438 | 0.2188 | 0.2618 |
| **Pubmed-BART** | 10.80 | 0.3789 | 0.3242 | 0.3426 | 0.1027 | 0.0875 | 0.0926 | 0.3464 | 0.2963 | 0.3134 |
| **Pubmed-BART** | | | | | | | | | | |
| **- w one citation** | 11.51 | **0.3816** | 0.3288 | 0.3461 | **0.1070** | 0.0926 | 0.0971† | **0.3473** | 0.2998 | 0.3154 |
| **- w citation agg.** | **10.54** | 0.3758 | **0.3427**† | **0.3522**† | 0.1039 | **0.0946**† | **0.0973**† | 0.3450 | **0.3143**† | **0.3233**† |

Table 2: Automatic evaluation based on ROUGE scores. LEAD-3, ORACLE, and ChatGPT were excluded from the performance comparisons as they were not trained on the datasets. However, we use them as reference models to provide insights into the potential performance achievable on our datasets. For each metric, the best overall score is highlighted in **bold**, and the baseline score is underlined. ↑ / ↓ indicates the higher/lower the better, respectively. **- w one citation** to denote the input configuration where the document is composed with one randomly selected citation abstract. **- w citation agg.** denotes our proposed citation abstract aggregation framework. † denotes that the citation-enhanced model results are statistically significant with respect to the base model (Pubmed-BART) by way of Mann-Whitney U test.

ploy BERTScore (Zhang et al., 2019) (BeS) and BartScore (Yuan et al., 2021) (BaS) to assess the quality of the generated outputs. We also introduce some readability metrics, e.g. Flesch-Kincaid (FLK) and Coleman-Liau Index (CLI), to evaluate the readability of the generated text. This comprehensive evaluation allows for a more robust assessment of the summarisation performance across multiple dimensions.

## 5.2 Implementation Details

All of the pre-trained models used are restored from the publicly available checkpoints on Hugging Face[9]. The checkpoints we selected include: LED[10], PEGASUS[11], BART[12], Pubmed-LED[13], Pubmed-PEGASUS[14], and Pubmed-BART[15].

To make the comparison fair, all input text is chunked according to the minimal input size limitation of selected language models. In our experiments, it is BART (1024 tokens). Models are trained for up to 10 epochs on a Tesla A40 machine, which has 40 GB GPU memory, and the best checkpoints are kept based on the perplexity of generated responses during validation for the

| Model | Referenced | | Unreferenced | |
|---|---|---|---|---|
| | BeS↑ | BaS↑ | FLK↓ | CLI↓ |
| **Golden** | 100 | -0.27 | 16.49 | 15.69 |
| **ChatGPT** | 85.56 | -3.11 | 16.19 | 16.31 |
| **Pubmed-LED** | 82.85 | -3.37 | 16.33 | 15.31 |
| **Pubmed-PEGASUS** | 81.97 | -3.42 | 15.78 | 14.52 |
| **Pubmed-BART** | 83.34 | -3.36 | 14.45 | 13.32 |
| **Pubmed-BART** | | | | |
| **- w one citation** | 83.34 | -3.36 | 14.78 | 13.51 |
| **- w citation agg.** | **83.60** | **-3.35** | **13.82** | **13.20** |

Table 3: Automatic evaluation on more metrics. **BeS** and **BaS** denote the F1 values of BERTScore and BartScore, respectively. **FLK** and **CLI** denote the readability scores of Flesch-Kincaid and Coleman-Liau Index, respectively.

generation on the testset. The batch size is set to 16, and the learning rate is $1e^{-4}$, with the Adam optimizer selected for training. For more details, please refer to the Appendix A.1.

## 5.3 Automatic Evaluation

The results of all experiments are presented in Table 2. It can be observed that our proposed framework (**-w citation agg.**) significantly outperforms all baseline models across all ROUGE scores (F1 scores), indicating substantial enhancements in the summarisation capability of biomedical papers. To be more specific, the incorporation of citation knowledge has contributed to a substantial improvement in recall, with ROUGE-1 exhibiting a 5.7% increase and ROUGE-2 demonstrating an 8.1% increase. This suggests that the integration of citation

[9] https://huggingface.co/models
[10] https://huggingface.co/allenai/led-base-16384
[11] https://huggingface.co/google/pegasus-x-base
[12] https://huggingface.co/facebook/bart-base
[13] https://huggingface.co/Blaise-g/led_pubmed_sumpubmed_1
[14] https://huggingface.co/google/pegasus-pubmed
[15] https://huggingface.co/mse30/bart-base-finetuned-pubmed

knowledge has facilitated the utilisation of more similar expressions extracted from the reference abstracts.

In addition, within our framework, the language model achieves substantially lower perplexity (PPL) and ROUGE-L scores, signifying an improvement in the language quality and reduced confusion during summary generation. We hypothesise that the decrease in PPL and ROUGE-L indicates that the language model has learned writing styles and relevant biomedical terminologies by referring to the abstracts of cited papers.

Regarding the ablation study, **-w one citation** yields a slight improvement compared to the baseline model Pubmed-BART but exhibits a higher perplexity. This observation suggests that the direct inclusion of random citation content may introduce certain noise. In contrast, our attention-based mechanism enables the neural networks to dynamically select and aggregate important information from multiple citations, effectively addressing confusion issues associated with additional inputs.

In Table 3, we present the results of additional evaluation metrics. BertScore and BartScore, as machine learning-based metrics, measure the semantic similarity between the generated summaries and the reference abstracts. Flesch-Kincaid and Coleman-Liau metrics assess text readability on a vocabulary level. Across all these metrics, **-w citation agg.** outperforms all baseline models, showcasing the advantages of introducing citation knowledge with our framework. Further analysis within the "Pubmed-BART" model reveals that using a single citation results in a slight decrease in BERTScore and BartScore, along with a slightly higher Flesch-Kincaid score (14.78) and Coleman-Liau Index score (13.51). However, employing citation aggregation leads to improvements across all metrics, with BERTScore (83.60), BartScore (-3.35), Flesch-Kincaid score (13.82), and Coleman-Liau Index score (13.20). This analysis confirms our initial hypothesis, that directly introducing a random citation may introduce noise that hampers model performance, while our aggregation model comprehensively considers all citation papers, effectively reducing the random noise introduced by a single citation.

## 5.4 Human Evaluation

In order to obtain a more comprehensive evaluation of the generated summaries, we also incorporate

| Score (1 to 5) | Human Evaluation | | | |
|---|---|---|---|---|
| | **Flu** | **Rea** | **Rel** | **Inf** |
| **Golden** | 4.86* | 4.70* | 5.0* | 5.0* |
| **ChatGPT** | 3.80 | 3.77 | 4.33* | 4.33* |
| **Pubmed-LED** | 2.70 | 2.67 | 3.83* | 3.70 |
| **Pubmed-PEGASUS** | 2.33 | 2.33 | 3.53* | 3.27 |
| **Pubmed-BART** | 2.73 | 2.73 | 3.77* | 3.57 |
| **Pubmed-BART** | | | | |
| **- w one citation** | 2.87 | 2.77 | 3.93* | 3.63 |
| **- w citation agg.** | **2.96** | **2.93** | **4.0*** | **3.83*** |

Table 4: Results of Human Evaluation. **Flu**, **Rea**, **Rel** and **Inf** denote Fluency, Readability, Relevance, and Informativeness, respectively. The best scores are in **bold**, and the second best are underlined. **ChatGPT** and **Golden** are not included in comparison. We calculate Fleiss' Kappa $\kappa$ for each metric. The majority of results demonstrate a moderate level of agreement ($\kappa \in (0.4, 0.6]$), and results with a higher level of agreement are marked with $*$.

human evaluation. This evaluation focuses on four key aspects: fluency, readability, relevance, and informativeness. Fluency assessment aims to measure the overall quality of the language used in the summaries. Readability evaluation determines the extent to which the summaries are easily understandable by readers. Relevance assessment examines whether the content of the summaries is pertinent and aligned with the content of the input document. Informativeness measurement evaluates the extent to which the generated summaries provide sufficient and meaningful information derived from the given input. By incorporating human evaluation, we can assess subjective aspects of summary quality that automated metrics may not fully capture.

Considering the difficulty of evaluating generated summaries, which requires a thorough understanding of the content in both the source papers and the summaries, it is imperative that human evaluators possess a strong background in academic writing and biomedical knowledge. We invite 3 qualified evaluators by snowball sampling to rate 30 randomly sampled instances from the testset. In order to minimise biases and increase inter-annotator agreement, the evaluators were provided with the same annotation guide (see Appendix A.3). The results of the human evaluation are presented in Table 4. It can be observed that both the - *w one citation* and - *w citation agg.* models exhibit superior performance compared to other baseline models, thereby affirming the effectiveness of our proposed framework.

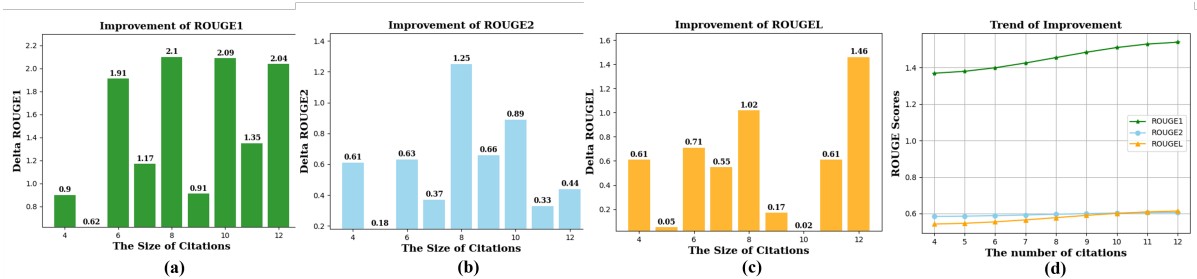

Figure 4: (a), (b) and (c) illustrate the bar charts depicting the performance enhancement achieved by the **-w citation agg.** method over the **PubmedBART** model. The improvement is calculated as the difference between the scores obtained by **-w citation agg.** and **PubmedBART**. In these bar charts, we report the ROUGE F1 scores. Additionally, Figure 1(d) exhibits the smoothed curve of the ROUGE scores data, obtained using Gaussian kernel smoothing technique. Due to the limit of GPU memory, we limit the input of the citation aggregation network to 12 citation papers.

To delve further into the evaluation, the metrics of Relevance and Informativeness underscore the improved capability to extract relevant information from the input content and generate comprehensive abstracts. Additionally, the fluency and readability metrics assess the language quality, indicating that the language model generates abstracts that are more coherent and natural. However, it is important to note that the tested Pretrained Language Models (PLMs) exhibited a notable disparity in language quality when compared to the performance of ChatGPT. This discrepancy can be attributed to the substantial difference in model size, with Chat-GPT having 130 billion parameters, whereas the tested PLMs have less than 5 billion parameters.

### 5.5   In-depth Analysis

To further investigate the impact of the citation knowledge aggregation module, we conduct an evaluation to assess the improvement in the generated abstracts. This evaluation involves comparing the performance of our proposed framework, denoted as **-w citation agg.**, against the base model **Pubmed-BART** using ROUGE scores. The results, presented in Figure 4 as (a), (b), and (c), illustrate the increase in ROUGE scores (F value) for different numbers of citations. The inclusion of citations is shown to have a positive effect on the abstract generation process. The Gaussian kernel smoothed increasing curve, depicted in Figure 4 (d), indicates a clear trend: as more citation abstracts are introduced, the language model exhibits greater improvements. The results highlight the potential of leveraging citation information to enhance the quality of generated abstracts.

## 6   Conclusion

In conclusion, we proposed a novel attention-based citation aggregation model that incorporates domain-specific knowledge from citation papers. By integrating this additional information, our model enables neural networks to generate summaries that benefit from both the paper content and the associated knowledge extracted from citation papers. Furthermore, we introduced a specialized biomedical summarisation dataset, which served as a valuable resource for evaluating and advancing our research. The effectiveness of our approach was demonstrated through extensive experiments, where our model consistently outperformed state-of-the-art methods in biomedical text summarisation. The results highlight the significant improvements achieved by leveraging knowledge from citation papers and the potential for our model to enhance the understanding of biomedical literature through natural language generation techniques.

### Acknowledgements

Chen Tang is supported by the China Scholarship Council (CSC) for his doctoral study (File No.202006120039). We also gratefully acknowledge the anonymous reviewers for their insightful comments.

### Limitations

In the field of text summarisation, two main approaches are commonly employed: extractive summarisation and abstractive summarisation. While extractive summarisation composes summaries by directly selecting sentences from the input content, abstractive summarisation generates summaries

that are not bound to the input content, providing greater flexibility but posing challenges in management and control. In this work, due to resource and time constraints, we focused on implementing an abstractive summarisation model and did not further conduct experiments to develop an extractive summarisation counterpart using our proposed algorithm. However, it is worth noting that our proposed approach has shown promising results, emphasizing the importance of leveraging citation papers to enhance the performance of language models in generating high-quality biomedical summaries. Theoretically, the aggregation of knowledge from citation papers can also be beneficial for extractive summarization approaches.

## Ethics Statement

Our new dataset is derived from an existing publicly available corpus released by the Allen Institute, which is a comprehensive biomedical literature corpus licensed under the Apache License 2.0. We have diligently adhered to the terms and conditions of the license and followed all provided instructions. Furthermore, we have familiarized ourselves with and acknowledged the ACM Code of Ethics and Professional Conduct[16]. We approach our professional responsibilities with utmost seriousness, ensuring that our study upholds ethical principles in every aspect.

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

## A  Appendices

### A.1  Implementation Details

**ChatGPT Prompts**  The performance of ChatGPT is highly reliable on the quality of input prompts. We manually design and test prompts of abstract summarisation, and select the best cases as the experimental results.

**Others**  The Gaussian kernel smoothing used in Figure 4 is implemented with the *gaussian_filter1d* function from the python package of *scipy.ndimage*. The ROUGE score evaluation is implemented with the python package *rouge_score*.  The readability scores such as Flesch-Kincaid (FLK) and Coleman-Liau Index (CLI), are implemented with the python package *py-readability-metrics*. BertScore is *bert_score*, and BartScore is from the GitHub repository of https://github.com/neulab/BARTScore.

### A.2  Automatic Evaluation

Table 6 shows the full results of BertScore and BartScore.

### A.3  Human Evaluation

In addition to automatic evaluation metrics, we conducted a comprehensive human evaluation to assess the quality of biomedical summarization generated by the different models. The human evaluation aimed to capture important aspects of summarization, including fluency, readability, and relevance.

For the human evaluation, we recruited a group of expert annotators with a strong background in biomedical research. The annotators were provided with a set of summaries generated by each model and were asked to rate them on a Likert scale ranging from 1 to 5. The Likert scale allowed annotators to provide a subjective assessment of the summaries based on their expertise and judgment. The four aspects evaluated in the human evaluation were as follows:

**Fluency**: Annotators assessed the language quality and coherence of the summaries. They considered factors such as grammar, sentence structure, and overall fluency of the generated text. Higher ratings on the Likert scale indicated better fluency.

**Readability**: Evaluators focused on the readability and comprehensibility of the summaries. They assessed whether the generated summaries were clear, concise, and understandable to a non-expert audience. Higher ratings indicated better readability.

**Relevance**: An important criterion was the relevance of the summaries to the original input documents. Annotators evaluated whether the summaries captured the main ideas, key findings, and important concepts present in the source documents. Higher ratings indicated greater relevance.

**Informativeness**: Evaluate the extent to which the generated summaries provide sufficient and meaningful information derived from the given input. Assess the comprehensiveness and completeness

| Index | Target | Template |
|-------|--------|----------|
| **1** | Generate Summaries | Write a summary according to given paper content, which is part of a medical scientific paper (A). The length of generated summary is expected to be larger than 130 words and less than $\{MAX_ABS_LEN\}$ words. $\backslash n \backslash n$ The paper content of (A) is: $\{DOC\}$ $\backslash n \backslash n$ Output: |
| **2** | Generate the Human Evaluation Guideline | Write a detailed humam evaluation guideline based on the following content: $\{X\}$ |

Table 5: The examples of ChatGPT prompt templates. The variable in the brackets { } will be replaced by the actual text during the utilisation.

| Model | BerS-P↑ | BertS-R↑ | BertS-F1↑ | BartS↑ |
|-------|---------|----------|-----------|--------|
| **Golden** | 100 | 100 | 100 | -0.26 |
| **ChatGPT** | 86.46 | 84.71 | 85.56 | -3.11 |
| **Pubmed-LED** | 82.91 | 82.83 | 82.85 | -3.37 |
| **Pubmed-PEGASUS** | 82.55 | 81.43 | 81.97 | -3.42 |
| **Pubmed-BART** | 84.26 | 82.49 | 83.35 | -3.38 |
| Pubmed-BART | | | | |
| - w one citation | 84.20 | 82.54 | 83.34 | -3.36 |
| - w citation agg. | **84.33** | **82.92** | **83.60** | **-3.35** |

Table 6: **BeS** and **BaS** denote of BERTScore and BartScore, respectively. **P**, **R**, **F1** represents the precision, recall and F1 values, respectively.

of the summary. Consider the inclusion of important details and relevant facts.

By utilising a Likert scale with a range of 1 to 5, we were able to capture nuanced evaluations from the annotators. This human evaluation provided valuable insights into the overall performance of the models from the perspectives of fluency, readability, and relevance, allowing us to gain a deeper understanding of their summarization capabilities in the biomedical domain.

### A.4 Future Works

In this paper, we propose a novel framework designed to enhance the performance of biomedical text summarisation using pre-trained language models. Recent years have witnessed the emergence of increasingly potent open-source language models, exemplified by Llama 2[17] and Baichuan[18]. However, the practical implementation of our approach on these immensely large-scale models has been constrained by high computational demands. Consequently, we anticipate the need for more advanced GPU hardware or optimised models, such as distilled models (Yang et al., 2023b), to render the training of these models feasible. Presently, there are two primary ways for advancing our research in citation network-enriched text summarisation:

and we have to wait for more advanced GPU devices or more optimised models (e.g. distilled models) to make training those models to be practical. Currently, there are two main direction to further improve our citaion networks enhanced text summarisation: (1) The development of a more efficient neural network that can effectively incorporate the graph-based features derived from citations (Tang et al., 2023a; Yang et al., 2023a). (2) The identification and extraction of key information from both the input document and its associated citations to enhance language understanding (Huang et al., 2022; Tang et al., 2022c). We defer the exploration of these directions to future research endeavors.

---

[17]https://ai.meta.com/llama/.
[18]https://github.com/baichuan-inc/Baichuan2