# OpenReview forum: "Improving Biomedical Abstractive Summarisation with Knowledge Aggregation from Citation Papers"
_EMNLP/2023/Conference — EMNLP 2023 Main_

### Official Review · Reviewer_u9sN · 2023-07-21

**Soundness:** 4

**Excitement:**

3: Ambivalent: It has merits (e.g., it reports state-of-the-art results, the idea is nice), but there are key weaknesses (e.g., it describes incremental work), and it can significantly benefit from another round of revision. However, I won't object to accepting it if my co-reviewers champion it.

**Paper Topic And Main Contributions:**

The authors propose a large-scale dataset to improve the state of abstractive summarization in biomedical domain. In particular they use the abstracts of citations of an article and the source article and propose a novel aggregation strategy to efficiently utilize the knowledge of the citation network and then generating refernce summary as the absract.

**Questions For The Authors:**

Please answer the queries stated in Reason to Reject section.

**Reasons To Accept:**

1. a large-scale dataset is a good contribution to summarization space especially in biomedical domain where such large scale datasets are very sprasely available

2. an interesting approach to enhance the knowledge learnt for decoding step to produce better relevant abstract.

3. an in-depth analysis and wide variety of baselines considered in the study to show improvements using the proposed strategies.

**Reasons To Reject:**

1. although the choice of models seems fine at first, but I am not sure as to how much of the citation information is actually being utilized. The maximum input size can only be 1024 and given the size of articles and the abstracts I am not sure how much inforamtion is being used in either. Can you comment on how much information is lost at token-level that is being fed to model.

2. I like the idea of aggregation using various cited articles. The only problem and I might be possibly confused as to how are you ensuring the quality of chosen articles. It could be so that the claims made in the article might also be contradicting to the claims made in cited articles or might not at all be related to the claims being discussed in the articles. Do you have any analysis on correlation between cited articles and the main articles, and whether it affects the quality of generation?

3. The authors have reprt significane testing but I think the choice of test might be incorrect. Since the comparision is to be done between two samples generated from same input why not some paired test setting was used like wilcoxon signed ranked test?

**Reproducibility:**

4: Could mostly reproduce the results, but there may be some variation because of sample variance or minor variations in their interpretation of the protocol or method.

**Reviewer Confidence:**

2: Willing to defend my evaluation, but it is fairly likely that I missed some details, didn't understand some central points, or can't be sure about the novelty of the work.

---

> ### Author Rebuttal · Authors · 2023-08-29
>
> Thanks for your detailed and insightful comments and we present our responses below.
>
> Reject_1: We would like to clarify that in our approach, for each citation paper, we concatenated its abstract with the original article and then fed it to the encoder for producing an encoding (i.e., for N citations papers, there will be N encodings produced). The features of the resulting encodings of the concatenated inputs are aggregated into a fixed size vector for the decoding process, as shown in Equations (3) and (4). Therefore, the input sequence is fixed and does not grow linearly with the number of citation papers. To be more specific, the length limit ratio of the content of the original article and the abstract of one citation paper to the input is set by 3 to 1 in our implementation, where 80% paper abstracts in the corpus have the words less than the limit and can be fully incorporated. We will make sure to make this clear in a final paper version by providing more implementation details in the appendix. We are happy to provide further info as the reviewer needs..
>
> Reject_2: We thank the reviewer for raising this interesting point. In our methodology, we aim to ensure the quality of the cited article information that is utilised during generation via the incorporation of the attention-based mechanism as described from line 242 to 270. This mechanism essentially assigns a score to each cited article in the knowledge aggregation process and, by training it in an end-to-end fashion, the idea is that this component will learn to capture aspects such as the correlation between the source and citation article. We believe that the improvements demonstrated across both automatic and human evaluations suggest the component can effectively  learn such information, but also  agree that a more explicit modelling of such  relationships between the source and citation articles could offer further benefits to the knowledge aggregation process. We leave this to our future work.
>
> Reject_3: Thanks for your suggestion! We would like to clarify the input (having citation information) of models "- w one citation" and "- w citation agg." are actually different from that (having no citations) of the baseline "Pubmed-BART", so we employ the Mann-Whitney U test rather than the Wilcoxon signed-rank test. We are happy to provide further info as the reviewer needs.

---

### Official Review · Reviewer_bXTQ · 2023-08-05

**Soundness:** 4

**Excitement:**

3: Ambivalent: It has merits (e.g., it reports state-of-the-art results, the idea is nice), but there are key weaknesses (e.g., it describes incremental work), and it can significantly benefit from another round of revision. However, I won't object to accepting it if my co-reviewers champion it.

**Paper Topic And Main Contributions:**

This paper incorporates the abstracts of cited papers into the content of source articles for biomedical literature summarization (CORD-19). Specifically, they construct a biomedical literature dataset, which contains the citation graphs and content of both cited and source papers; they revised the attention mechanism during the encoding process to measure the importance between original source paper content and abstracts of cited papers; they conducted a series of experiments to evaluate the performance of the proposed methods. The results show that their approach was able to have better performance than the baseline models.

**Questions For The Authors:**

A. It would be helpful if the authors could elaborate more on the lower precision scores of the best-performing method.

B. Could the authors discuss more on how the longer sequences are handled during the encoding process?

C. As the source articles could cite related work for different purposes (e.g., discussing background information, comparing the method, using the dataset, etc). Semantic Scholars are also adding this citation function information to their database. Due to the different purposes for the citation, the relevance or importance of the cited articles might be different in regard to the source articles. I wonder if adding such citation function information to the encoders would also help to decide the importance of the cited papers?




**Reasons To Accept:**

Biomedical text summarisation is an important one to the NLP community. Developing methods to improve the current models would be potentially beneficial for many downstream applications in this field.

It is a nice idea of including external knowledge from the abstracts of cited articles.

The developed dataset would be useful for the community.

**Reasons To Reject:**

Combining the cited abstracts and original content of source papers together could make the input sequence very long for the encoders. It is not very clear how the longer sequences are handled during the encoding process, which might affect how the external knowledge from the cited papers is utilized.

Based on the reported results, the precision scores of the best-performing model (citation agg) seem to be lower compared to other methods across the three evaluation methods (as reported in Table 2).

**Reproducibility:**

3: Could reproduce the results with some difficulty. The settings of parameters are underspecified or subjectively determined; the training/evaluation data are not widely available.

**Reviewer Confidence:**

3: Pretty sure, but there's a chance I missed something. Although I have a good feel for this area in general, I did not carefully check the paper's details, e.g., the math, experimental design, or novelty.

**Typos Grammar Style And Presentation Improvements:**

line 321: are unable -? able?

line 336-337: we were able to use...as the base language model to fine-tune on our dataset: do you mean unable to do so? as ChatGPT is close-source?

Table 2 - w citeion agg: w citation agg

I suggest the authors include the literature review section to the main content of the paper.

---

> ### Author Rebuttal · Authors · 2023-08-29
>
> Thanks for your detailed and insightful comments and we present our responses below.
>
> Reject_1: In our approach, for each citation paper, we concatenate its abstract with the original article and then feed it to the encoder to produce an encoding (i.e., for N citations papers, there will be N encodings produced). The resulting encodings are then aggregated into a fixed size vector for the decoding process, as shown in Equations (3) and (4). Therefore, the input sequence is fixed and does  not grow linearly with the number of citation papers. We will make sure to make this more clear in our final paper version.
>
> Reject_2: The lower precision scores  might potentially be attributed to a few factors .   First of all, the summaries generated by the baselines tend to contain more repeated words used in the ground truth leading to higher precision, but the actual quality of the summary is not higher, as evidenced by the lower scores for metrics of readability, bert/bart scores, and human evaluation in Table 2 and 3. In addition, our model incorporates more novel external phrases sourced from the citation papers, which have not been used in the ground truth, and hence has some impact on the precision.
>
> Question_A: Same as Reject_2 .
>
> Question_B: Same as Reject_1. In our approach, we conceptualise these citations as discrete nodes within a citation graph, thereby ensuring that the length of the input sequence remains invariant as the number of nodes (i.e., citations) increases.
>
> Question_C: This is a very good suggestion, and we believe the citation function information will be a very useful source of information. For instance, we could consider placing more emphasis on the citations whose purpose is for discussing background information, which will be more useful for feature enhancement. We will explore this in our future work.
>
> Typos Grammar Style And Presentation Improvements: All has been done, and we will put the literature section to the main content given an extra page allowed in the final version.

---

### Official Review · Reviewer_AoVu · 2023-08-12

**Typos Grammar Style And Presentation Improvements:** N/A
**Soundness:** 5

**Excitement:**

4: Strong: This paper deepens the understanding of some phenomenon or lowers the barriers to an existing research direction.

**Missing References:**

N/A

**Paper Topic And Main Contributions:**

This paper has 2 major contributions. Firstly, it has introduced a large scale biomedical research article summarization corpus. Secondly, it has proposed a novel approach to improve the performance of the LLMs for biomedical research article summarization. It's true that research articles can't express the whole thought in a standalone manner due to page limit restriction. To support the claims, they need to use citations. And to get proper knowledge enriched summaries, it is required to have the background knowledge which can be obtained via the citation network. This paper has explored this aspect fully and they have achieved state-of-the-art performance.

**Reasons To Accept:**

This is a great work which reflects the idea that for scientific document summarization, the model should have some background knowledge, specially for biomedical documents. And incorporating the abstracts of the cited papers it provides enriched features (background information) for the LLMs. Furthermore, the authors have introduced a large corpus for biomedical document summarization. Furthermore, they have achieved the state-of-the-art performance,

**Reasons To Reject:**

From my side there is no reason to reject this paper.

**Reproducibility:**

5: Could easily reproduce the results.

**Reviewer Confidence:**

4: Quite sure. I tried to check the important points carefully. It's unlikely, though conceivable, that I missed something that should affect my ratings.

---

> ### Author Rebuttal · Authors · 2023-08-29
>
> We are extremely grateful for your thoughtful and positive review of our paper. We also appreciate your detailed evaluation and your recognition of the contributions and merits of our work.

---

### Meta-Review · Area_Chair_cen5 · 2023-09-19

**Recommendation:** 5

**Metareview:**

In this paper, the authors propose a method to generate summaries using abstracts of citations of an article and the source article by the proposed aggregation strategy based on the graph-styled citation network. They also create a large-scale dataset for abstractive summarization in the biomedical domain. The experimental results on their created dataset show that their proposed method outperformed baselines. The authors answered reviewers' questions regarding input to the model and cited articles in detail. After the rebuttal, the reviewers agreed with their score towards accepting the paper; thus, there is no reason to unfollow their decision.

---

### Decision · Program_Chairs · 2023-10-07

**Decision:**

Accept-Main

**Comment:**

In this paper, the authors propose a method to generate summaries using abstracts of citations of an article and the source article by the proposed aggregation strategy based on the graph-styled citation network. They also create a large-scale dataset for abstractive summarization in the biomedical domain. The experimental results on their created dataset show that their proposed method outperformed baselines. The authors answered reviewers' questions regarding input to the model and cited articles in detail. After the rebuttal, the reviewers agreed with their score towards accepting the paper; thus, there is no reason to unfollow their decision.